# *NUDT21*-spanning CNVs lead to neuropsychiatric disease and altered MeCP2 abundance via alternative polyadenylation

Vincenzo A Gennarino[1,2†], Callison E Alcott[2,3,12†], Chun-An Chen[1,2], Arindam Chaudhury[4,5], Madelyn A Gillentine[1,2], Jill A Rosenfeld[1], Sumit Parikh[6], James W Wheless[7], Elizabeth R Roeder[1,8], Dafne DG Horovitz[9], Erin K Roney[1], Janice L Smith[1], Sau W Cheung[1], Wei Li[10], Joel R Neilson[4,5], Christian P Schaaf[1,2]*, Huda Y Zoghbi[1,2,3,8,11]*

[1]Department of Molecular and Human Genetics, Baylor College of Medicine, Houston, United States; [2]Jan and Dan Duncan Neurological Research Institute, Texas Children's Hospital, Houston, United States; [3]Program in Developmental Biology, Baylor College of Medicine, Houston, United States; [4]Department of Molecular Physiology and Biophysics, Baylor College of Medicine, Houston, United States; [5]Dan L. Duncan Cancer Center, Baylor College of Medicine, Houston, United States; [6]Center for Child Neurology, Cleveland Clinic Children's Hospital, Cleveland, United States; [7]Department of Pediatric Neurology, Neuroscience Institute and Tuberous Sclerosis Clinic, Le Bonheur Children's Hospital, University of Tennessee Health Science Center, Memphis, United States; [8]Department of Pediatrics, Baylor College of Medicine, Houston, United States; [9]Depto de Genetica Medica, Instituto Nacional de Saude da Mulher, da Criança e do Adolescente Fernandes Figueira, Rio de Janeiro, Brazil; [10]Division of Biostatistics, Dan L Duncan Cancer Center, Department of Molecular and Cellular Biology, Baylor College of Medicine, Houston, United States; [11]Howard Hughes Medical Institute, Baylor College of Medicine, Houston, United States; [12]Medical Scientist Training Program, Baylor College of Medicine, Houston, United States

*For correspondence:
hzoghbi@bcm.edu (HYZ);
schaaf@bcm.edu (CPS)

†These authors contributed equally to this work

Competing interests:
See page 10

**Abstract** The brain is sensitive to the dose of MeCP2 such that small fluctuations in protein quantity lead to neuropsychiatric disease. Despite the importance of MeCP2 levels to brain function, little is known about its regulation. In this study, we report eleven individuals with neuropsychiatric disease and copy-number variations spanning *NUDT21*, which encodes a subunit of pre-mRNA cleavage factor Im. Investigations of *MECP2* mRNA and protein abundance in patient-derived lymphoblastoid cells from one *NUDT21* deletion and three duplication cases show that *NUDT21* regulates MeCP2 protein quantity. Elevated *NUDT21* increases usage of the distal polyadenylation site in the *MECP2* 3′ UTR, resulting in an enrichment of inefficiently translated long mRNA isoforms. Furthermore, normalization of *NUDT21* via siRNA-mediated knockdown in duplication patient lymphoblasts restores MeCP2 to normal levels. Ultimately, we identify *NUDT21* as a novel candidate for intellectual disability and neuropsychiatric disease, and elucidate a mechanism of pathogenesis by MeCP2 dysregulation via altered alternative polyadenylation.

**eLife digest** The X-chromosome carries a number of genes that are involved in a child's intellectual development. One of these genes encodes a protein called MeCP2, which is important for brain function after birth. Mutations in the *MECP2* gene cause a disorder known as Rett syndrome. At around 18 months of age, affected children begin to lose the cognitive and motor skills that they had previously acquired. Individuals with extra copies of this gene also show cognitive impairments. For both diseases, individuals with levels of the MeCP2 protein that are the most different from those found in healthy individuals also show the most severe symptoms.

To produce the protein that is encoded by a particular gene, enzymes inside the cell must first make a copy of that gene using a molecule called messenger ribonucleic acid (or mRNA). This mRNA is then used as a template to assemble the protein itself. In the case of *MECP2*, two different mRNA templates are produced: a long version and a short version. A gene called *NUDT21* makes a protein that regulates whether the long or short version of *MECP2* mRNA is made.

Gennarino, Alcott et al. have now discovered that people with too many, or too few, copies of the *NUDT21* gene have intellectual disabilities and altered levels of MeCP2 protein. Specifically, individuals with extra copies of *NUDT21*—and thus higher levels of the corresponding protein—produce more of the long *MECP2* mRNA. The production of proteins from this long mRNA is less efficient than from the short mRNA; therefore, these individuals have lower levels of MeCP2 protein. The opposite is true for individuals who lack a copy of the *NUDT21* gene.

To confirm these data, Gennarino, Alcott et al. grew cells in the laboratory from patients with extra copies of the *NUDT21* gene and found that reducing the production of its protein returned the levels of the MeCP2 protein back to normal. These findings show that alterations in the *NUDT21* gene cause changes in the level of MeCP2 protein in cells and leads to neuropsychiatric diseases.

## Main text

Methyl CpG-binding protein 2 (MeCP2) binds methylated cytosines (*Lewis et al., 1992*, *Guo et al., 2014*) and affects the expression of thousands of genes (*Chahrour et al., 2008*). *MECP2* loss-of-function is the predominant cause of Rett syndrome, a postnatal neurological disorder that typically manifests around 18 months of age with developmental regression (*Amir et al., 1999*; *Ravn et al., 2005*; *Pan et al., 2006*). *MECP2* mutations additionally cause other neurological disorders, such as non-specific autism (*Carney et al., 2003*), Angelman-like syndrome (*Imessaoudene et al., 2001*), and intellectual disability (*Orrico et al., 2000*). And in the absence of *MECP2* mutations, decreased MeCP2 expression has been detected in the brains of patients with autism, Angelman syndrome, and Prader–Willi syndrome (*Samaco et al., 2004*; *Nagarajan et al., 2006*).

Duplications within Xq28 involving *MECP2* account for 1% of X-linked intellectual disability (*del Gaudio et al., 2006*; *Friez et al., 2006*; *Lugtenberg et al., 2006*; *Meins et al., 2005*; *Van Esch et al., 2005*; *Cox et al., 2003*) and triplications encompassing *MECP2* lead to a more severe phenotype (*del Gaudio et al., 2006*). Mouse studies have established that *MECP2* alone within the duplicated and triplicated region is sufficient to cause all the neurological phenotypes of the duplication and triplication syndromes (*Collins et al., 2004*; *Samaco et al., 2012*). Notably, even small changes in MeCP2 protein level lead to neurocognitive deficits and behavioral abnormalities, and the severity of the phenotype correlates with the level of MeCP2 (*Chao and Zoghbi, 2012*).

*MECP2* is distinctive for its long 3′ UTR of about 8.5 kb that contains two predominant poly-adenylation (p(A)) sites: a proximal p(A) site located just after the last exon, and a distal p(A) site approximately 8 kb from the final exon, which result in short and long messenger ribonucleic acid (mRNA) isoforms, respectively (*Coy et al., 1999*; *Shahbazian et al., 2002*). 3′ UTRs are important for fine-tuning transcript and protein levels because they contain binding sites for regulatory molecules such as RNA-binding proteins and microRNAs (miRNAs), (*Bartel, 2009*, *Gennarino et al., 2012*; *Gennarino et al., 2015*) which induce degradation of the mRNA or inhibit its translation. Thus, long isoforms allow for greater regulation due to an increased number of regulatory elements. The long 3′ UTR of *MECP2* harbors more than 50 putative miRNA-binding sites, including the miRNAs known to bind *MECP2*—miR-483-5p, miR-132 and miR-155—and reduce MeCP2 protein abundance (*Klein et al., 2007*; *Kuhn et al., 2010*; *Han et al., 2013*). The proximal p(A) site of *MECP2* is increasingly used throughout postnatal development, resulting in more

short mRNA isoforms and a concomitant increase in protein (*Balmer et al., 2003*), which may be driven by the loss of regulatory binding sites in its 3′ UTR. These observations highlight the complex and precise regulation of MeCP2 in the human brain.

Pre-messenger RNA cleavage factor Im (CFIm) regulates 3′ UTR length by mediating alternative poly-adenylation (APA) (*Millevoi and Vagner, 2010*). It binds pre-messenger mRNA as a hetero-tetramer that consists of a *NUDT21*-encoded CFIm25 dimer and any two of the paralogs CFIm59 or CFIm68 (or its splice variant, CFIm72) (*Kim et al., 2010*; *Yang et al., 2011*). In vitro, CFIm25 or CFIm68 knockdown leads to a transcriptome-wide increase in proximal APA site usage, whereas knockdown of other p(A) proteins, such as CFIm59, CPSF, CSTF, or CFIIm, has minimal effects on APA (*Gruber et al., 2012*; *Masamha et al., 2014*). Several models have been proposed to explain the role of CFIm25 in APA, but the exact mechanism remains unknown. A total of 59 genes have increased proximal p(A) site usage when comparing low- to high-*NUDT21* expressing glioblastoma tumors, and of these, 24 show the same effect as that following CFIm25 knockdown in HeLa cells. *MECP2* is the most affected, and its protein levels increase accordingly (*Masamha et al., 2014*). Moreover, Clip-seq data show enriched CFIm25 binding of the *MECP2* 3′ UTR near its p(A) sites (*Gruber et al., 2012*). Based on this, we hypothesized that CFIm25 regulates MeCP2 in humans by changing the ratio of short to long mRNA isoforms and that *NUDT21* gain- or loss-of-function will be correlated with neuropsychiatric disease.

To support our hypothesis, we sought patients with copy-number variation (CNV) of *NUDT21*. We identified eight individuals with duplications and three with deletions spanning *NUDT21*, all of whom had undergone clinical chromosome microarray testing. Of these, five consented to provide a detailed medical history (*Figure 1* and *Table 1*), and four provided blood samples for molecular studies (three with *NUDT21*-spanning duplications and one with a deletion).

Lymphoblastoid cell lines are commonly used in patient-specific molecular and functional studies of neurological disease (*Sie et al., 2009*). We, therefore, established immortalized lymphoblastoid cell lines of the four individuals and tested them for relative changes in MeCP2 protein and mRNA levels, and APA site usage. As hypothesized, in the duplication patients, we found ~60% more CFIm25 and ~50% less MeCP2 protein (*Figure 2A*), while in the deletion patient, we found ~50% less CFIm25 and ~50% more MeCP2 protein (*Figure 2B*), when compared to 13 age-matched control subjects. In order to prove that *NUDT21* alone can regulate MeCP2 levels, we performed an siRNA-mediated knockdown of *NUDT21* and assessed for changes in MeCP2 protein abundance. We found that *NUDT21* knockdown increases MeCP2 in patient-derived lymphoblastoid cells, and that reducing it to wild-type levels in the *NUDT21*-duplication patients rescues MeCP2 protein levels to that of the healthy controls (*Figure 2C*). To our knowledge, this is the first time that patient-derived lymphoblastoid cells have been transfected or transduced with siRNAs and that the functional consequence of a candidate gene within a larger CNV has been validated in vitro.

We then set out to understand how *NUDT21* CNVs lead to altered MeCP2 protein levels by investigating their effects on *MECP2* mRNA. Quantitative RT-PCR showed that total *MECP2* mRNA was elevated in both the duplication cases and the deletion patient (*Figure 3A*). However, using primer pairs that detected only the long 3′ UTR isoform of *MECP2*, we found that duplication patients have increased long *MECP2* (*Figure 3A*), indicating that elevated CFIm25 increases distal APA site usage. Conversely, the deletion patient showed a decrease in the long *MECP2* isoform, further indicating that *NUDT21* levels correlate with *MECP2* 3′ UTR length (*Figure 3A*). Northern blot analysis of RNA from the lymphoblastoid cells confirmed the switching between long and short *MECP2* isoforms for both duplication and deletion patients (*Figure 3—figure supplement 1*). Our analysis of the *MECP2* mRNA levels in *NUDT21*-spanning duplication and deletion subjects shows that CFIm25 regulates the ratio of short to long mRNA isoforms (*Figure 3A*).

To clarify how an increase in *MECP2* mRNA may still lead to decreased MeCP2 protein in the *NUDT21*-duplication patients, we performed a polyribosome fractionation assay to assess translation efficiency of total and long *MECP2* mRNA (*Figure 3—figure supplement 2*). We found that the *NUDT21*-duplication patients translate *MECP2* less efficiently than healthy controls, with a ~50% reduction in total *MECP2* mRNA in the polyribosome fraction and a dramatic enrichment of the long isoform in the non-polyribosome fraction (*Figure 3B*). These data support the hypothesis that *NUDT21*-duplication patients have decreased MeCP2 protein despite increased mRNA because *NUDT21* promotes distal APA site usage and the elevated RNA pool consists largely of the inefficiently translated long *MECP2* isoform (*Figure 3C*).

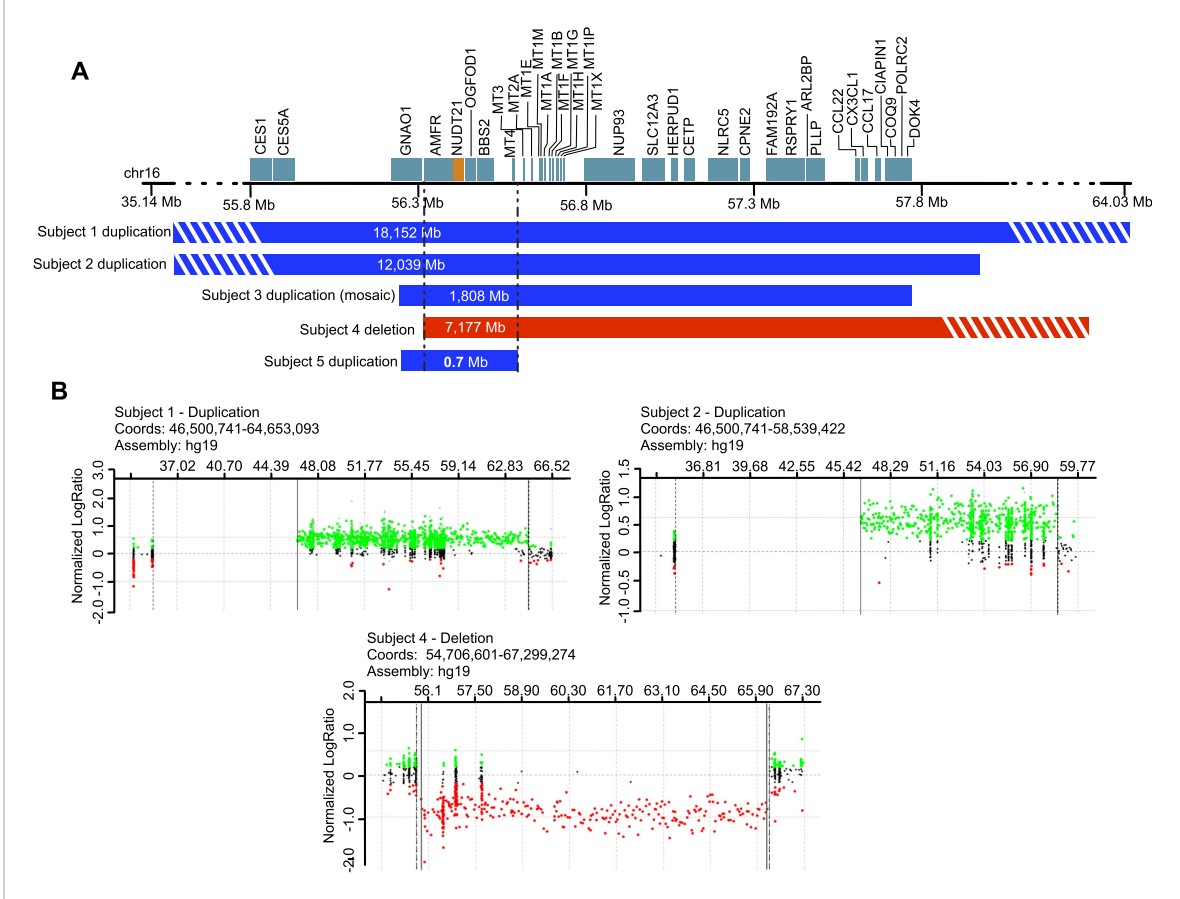

**Figure 1**. Subjects with *NUDT21*-spanning copy-number variations (CNVs). (**A**) Five intrachromosomal rearrangements of chromosome 16q including the *NUDT21* gene, identified by clinical array comparative genomic hybridization. Duplications shown in blue, deletion in red. Del, deletion; Dup, duplication; Mb, megabases. The striped bars indicate that the copy variant is not drawn to scale. (**B**) Array plots of oligonucleotide arrays on subjects 1, 2, and 4. The array plots of subject 3 and 5 were unavailable due to the closure of the Signature Genomics microarray laboratory in June 2014. Black dots indicate probes with normal copy-number, green dots indicate copy-number gain, and red dots indicate copy-number loss. Solid and dotted lines respectively define the minimum and maximum expected boundaries of the CNVs.

The CFIm25 subunit of pre-messenger RNA CFIm represents the first identified post-transcriptional protein regulator of MeCP2 and provides important insights into MeCP2 regulation during normal development and disease.

The presented cases of individuals with *NUDT21* CNVs and subsequent MeCP2 level changes suggest *NUDT21* as a novel candidate gene for intellectual disability and neuropsychiatric disease. Their CNVs, however, are non-recurrent, and all affect several genes in addition to *NUDT21,* so that the actual effect of the *NUDT21* dosage change alone on phenotype cannot be determined at this point. CNVs affecting only *NUDT21* or point mutation cases of *NUDT21* would provide important insight. However, we found no *NUDT21*-spanning CNVs in 30,466 healthy controls (*Pinto et al., 2007*; *Simon-Sanchez et al., 2007*; *Zogopoulos et al., 2007*; *International Schizophrenia C, 2008*; *Jakobsson et al., 2008*; *Itsara et al., 2009*; *Kirov et al., 2009*; *Shaikh et al., 2009*; *Conrad et al., 2010*; *International HapMap et al., 2010*; *Vogler et al., 2010*; *Banerjee et al., 2011*; *Campbell et al., 2011*; *Cooper et al., 2011*; *Genomes Project et al., 2012*), while among our group of 87,200 patients who underwent clinical chromosome microarray analysis, mostly for neurodevelopmental disorders, we identified 11 *NUDT21*-spanning CNVs (p = 0.025, one-tailed Chi square test without Yates correction), suggesting that CNVs involving *NUDT21* are indeed pathogenic. Moreover, of the four other genes common to all the patients' CNVs, only one is associated with neurological disease, *BBS2*, but it is autosomal recessive and is characterized by distinctive features not seen in the patients we studied.

**Table 1.** Molecular and clinical characteristics of four individuals with *NUDT21* duplication and one individual with *NUDT21* deletion

| | Subject 1 | Subject 2 | Subject 3 | Subject 4 | Subject 5 |
|---|---|---|---|---|---|
| Sex | F | M | M | F | F |
| Age | 10 years | 5 years | 15 years | 13 years | 8 years |
| Deletion/duplication | Duplication | Duplication | Duplication | Deletion | Duplication |
| Coordinates | chr16: 46,500,741-64,653,093 | chr16: 46,500,741-58,539,422 | chr16: 55,725,264-57,533,101 | chr16: 56,344,856-63,521,523 | chr16: 55,919,145-56,619,283 |
| Size | 18.15 Mb | 12.04 Mb | 1.81 Mb | 7.18 Mb | 0.70 Mb |
| Zygosity | Heterozygous | Heterozygous | Mosaic marker chromosome (present in 45% of cells) | Heterozygous | Heterozygous |
| Inheritance | De novo | De novo | De novo | Unknown | Not maternal* |
| Additional CNVs | 52 kb duplication of chr6: 152, 716, 341-152, 768, 421 (maternal) | None | None | 144 kb duplication of chr17: 427, 284-571, 275 (inheritance unknown) | None |
| Developmental delay/intellectual disability | Intellectual disability (clinical impression: moderate) | Significant developmental delay | Intellectual disability | Intellectual disability (full scale IQ 53) | Intellectual disability |
| Developmental regression | Yes, at 2 years of age | No | Yes | No | No |
| Autism spectrum disorder | Yes | No | Yes | No | Yes |
| Epilepsy | Single, isolated seizure at 5 years of age | No | No | Intractable, symptomatic partial onset seizures | No |
| ADHD | Yes | Yes | Unknown | Yes | No |

F, female; M, male; chr, chromosome; MB, megabases; CNVs, copy number variants; kb, kilobases; ADHD, attention deficit hyperactivity disorder.
*Father not available.

The clinical presentation of the three individuals with increased *NUDT21* copy number and decreased MeCP2 protein levels is not that of classic Rett syndrome, which is not surprising since individuals with Rett lack the protein in 50% of their cells, whereas the duplication of *NUDT21* causes a partial decrease in all cells. However, these individuals suffer from intellectual disability and autism spectrum disorder, and two of the three individuals manifested significant developmental regression, which is a rare clinical phenomenon and considered a hallmark of Rett syndrome. The identification of additional patients will help us better define the phenotypic overlap and differences between *NUDT21*-duplication and Rett syndrome, and for *NUDT21* deletion or loss-of-function and *MECP2* duplication syndrome.

The clinical and molecular data presented provide insight into the post-transcriptional regulation of *MECP2*, and identify *NUDT21* as a novel candidate for intellectual disability and neuropsychiatric disease. Further, we developed a new way of validating the role of individual genes within pathogenic CNVs by effectively transfecting patient-derived lymphoblastoid cells. Ultimately, this study provides an example of how the complexity of a CNV goes well beyond the affected chromosomal domain and the genes affected within. The better we understand gene networks, gene-protein, and protein–protein interactions, the better we will be positioned to identify the molecular bases of various neuropsychiatric disorders and design treatment strategies for the affected individuals.

## Materials and methods

### Human subjects

Of approximately 52,000 patients referred to the Baylor College of Medicine (BCM) Medical Genetics Laboratory (MGL) for clinical array comparative genomic hybridization (aCGH) analysis between April 2007 and February 2013, six probands with copy number variants affecting *NUDT21* were identified: five duplication cases and one deletion. We also screened the database of Signature Genomics

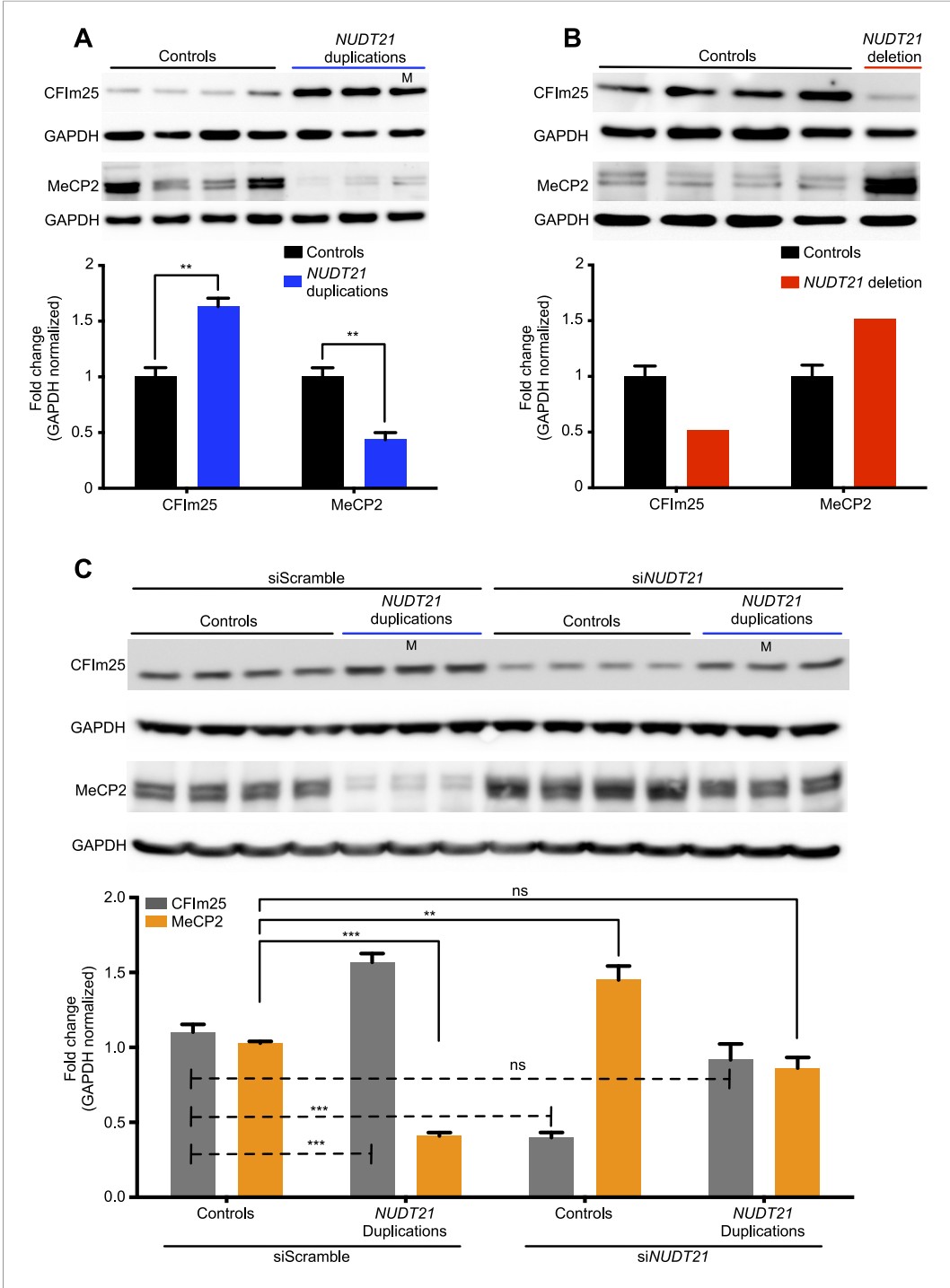

**Figure 2**. CFIm25 regulates MeCP2 protein levels in patient-derived lymphoblastoid cells with *NUDT21* CNVs. (**A**) Representative western blot picture for three duplication patients compared to four age-matched controls showing the increase of CFIm25 and decrease of MeCP2 protein levels. (**B**) Representative western blot picture for one deletion patient compared to four age-matched controls showing the decrease of CFIm25 and increase of MeCP2 protein levels. Quantification of protein levels for both CFIm25 and MeCP2 from three duplication patients and one deletion patient compared to a total of 13 age-matched controls are shown below the corresponding western blot. Data represent mean ± SEM from a total of six technical replicates. Data were normalized to GAPDH protein levels. (**C**) Western blot and its relative quantification showing that knockdown of *NUDT21* by siRNA-*NUDT21* nucleofection increases MeCP2 in control and duplication subjects, and normalizing CFIm25 in duplication patients rescue MeCP2 to control levels. Data represent mean ± SEM

*Figure 2. continued on next page*

*Figure 2. Continued*

from four age-matched control and three duplication cases. Data were normalized to GAPDH protein levels. **$p < 0.01$, ***$p < 0.001$. M, mosaic patient.

The following figure supplement is available for figure 2:

**Figure supplement 1**. siGLO nucleofection showing patient-derived lymphoblastoid cells can be transfected with small RNA.

Laboratories, based on approximately 35,200 samples submitted for clinical aCGH between November 2007 and February 2013, and identified three duplication cases and two deletions. The search was limited to copy-number variants <20 megabases. The clinical labs shared our contact information with the referring providers of all eleven cases, and we were subsequently contacted by five families who expressed interest in participating in this research study. Their copy-number variants had been detected by clinical aCGH on the following platforms: CMA-HR + SNP V9.1.1, Baylor College of Medicine (subject 1); Oligo V8.1.1, Baylor College of Medicine (subject 2); SignatureChipOS v2.0, Signature Genomics (subject 3); and Oligo V6.5, Baylor College of Medicine (subject 4). Following informed consent, approved by the Institutional Review Board for Human Subject Research at Baylor College of Medicine, we performed a comprehensive chart review of medical records and neuropsychological testing. A venous blood sample was provided by the probands in order to establish immortalized lymphoblastoid cell lines. All individuals with *NUDT21* copy-number variants were enrolled in a research study approved by the Institutional Review Board of Baylor College of Medicine (H-25466). The consent form specifically allows for sharing of medical information and physical exam findings. All individuals whose lymphoblastoid cell lines were enrolled as controls had been enrolled in a research study approved by the Institutional Review Board of Baylor College of Medicine (H-25531) as unaffected sibling controls.

## Lymphoblastoid cell culture

Venous blood of the recruited probands was drawn into ACD solution A tubes. Buffy coat was prepared, lymphocytes were pelleted, and transformed with Epstein–Barr virus and cyclosporin A following standard procedures. Cell lines were grown in RPMI 1640 medium (Invitrogen, Carlsbad, CA, United States) supplemented with 10% fetal bovine serum (Atlanta Biological, Flowery Branch, GA, United States) and 1% penicillin/streptomycin. Cell cultures were maintained at 37°C in a humidified incubator supplemented with 5% $CO_2$. Medium was renewed every 2 to 3 days to maintain the cell density between $1 \times 10^5$ and $2 \times 10^6$ cells/ml.

## Lymphoblastoid nucleofection

Lymphoblastoid cells were nucleotransfected with 300 nM of either Ambion small interfering RNA si*NUDT21* (s21771) or siScramble (4390843) using Amaxa 4D-Nucleofector and P3 Primary Cell 4D-Nucleofector X Kit (Lonza, Cologne, Germany, cat# V4XP-3024). Cells ($2 \times 10^7$) were centrifuged at 200×*g* for 10 min at room temperature, then resuspended in 100 µl P3 Primary Cell 4D-Nucleofector Solution with the added supplement. In microfuge tubes, 300 nM siRNA was mixed with the suspended cells and transferred to Nucleocuvette Vessels prior to nucleofection. Cells were nucleofected using the program FI-115 (efficiency). Post-nucleofection, cells were resuspended with 0.5-ml pre-warmed media (RPMI, 10% FBS, and 1% antibiotic–antimycotic) and transferred to T25 flasks. After 6 hr, medium was removed and replaced with fresh media. Cells were incubated at 37°C with 5% $CO_2$ for 48 hr prior to protein extraction and western blot analysis. The nucleofection protocol was optimized by transfecting $2 \times 10^7$ lymphoblastoid cells with 2 µM of siGLO Green transfection indicator (Dharmacon, Lafayette, CO, United States) or Control (empty) either using the program FI-115 (efficiency) or EO-115 (functionality). Cells were later incubated at 37°C with 5% $CO_2$ and collected at different time points from 12 to 72 hr for cytometric analysis using a BD LSR FORTESSA (BD Biosciences, San Jose, CA, United States) according to manufacture's instructions. We determined that the efficiency protocol was preferable because the nucleofection rate was higher, and the survival rate was acceptable. We extracted protein at 48 hr post-nucleofection because the survival was satisfactory, and it was the latest time point at which nucleofected small RNA was nearly 100%, which would allow the siRNA the most time to act (*Figure 2—figure supplement 1*).

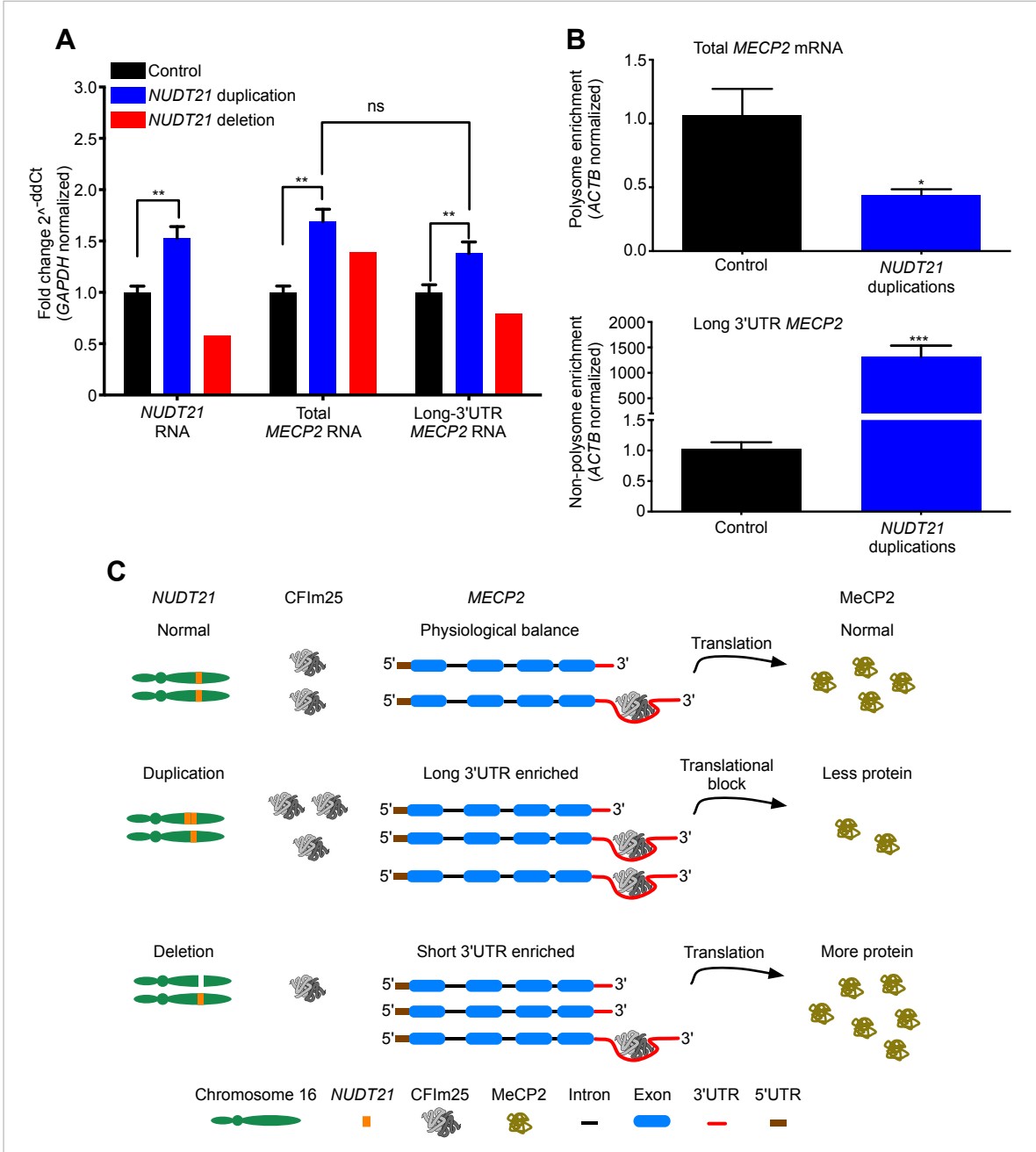

**Figure 3**. *NUDT21* mRNA levels correlate with inefficiently translated long *MECP2* mRNA. (**A**) RNA quantification by quantitative RT-polymerase chain reaction (qRT-PCR) from lymphoblastoid cells of *NUDT21* duplication and deletion patients. The bar graph shows the total mRNA fold change of *NUDT21*, total *MECP2*, and long *MECP2* for the three duplication patients and one deletion patient compared to 13 age-matched controls. Data represent mean ± SEM from five independent experiments. Data were normalized to *GAPDH* mRNA levels. (**B**) Relative polyribosomal and non-poyribosomal enrichment of total and long *MECP2* mRNA isoforms of *NUDT21* duplication patients compared to age-matched controls. Data represent mean ± SEM from a total of three control and duplication cases. Data were normalized to *ACTB* mRNA levels. (**C**) Proposed model showing that duplication and deletion patients have more or less CFIm25, respectively leading to a relative increase in long and short *MECP2* 3′ UTR isoforms. In both cases, there is an accumulation of mRNA: in the deletion patient, this leads to more MeCP2 protein, but in the duplication patients, it results in less MeCP2 protein due to a translational block from the CFIm25-mediated increase in long *MECP2* isoforms and putative binding of miRNAs or RNA-binding proteins to the 3′ UTR. *$p < 0.05$, **$p < 0.01$, ***$p < 0.001$.

The following figure supplements are available for figure 3:

**Figure supplement 1**. Northern blot assay from patient-derived lymphoblastoid cells.

**Figure supplement 2**. Polyribosome fractionation traces of control and duplication subjects.

## Western blot

Lymphoblastoid cell suspension cell cultures were collected at $6 \times 10^6$ confluence and processed for protein extraction. Cell pellets were lysed with modified RIPA buffer (180 mM NaCl, 0.5% NP-40, 0.5% Triton X-100, 2% SDS, and complete protease inhibitor cocktail (Roche, France)) by pipetting them up and down with a p1000 tip and then placed at room temperature rotisserie shaker for 10 min followed by boiling for 5 min. Then the lysates were repeatedly passed through a 27-G needle with a syringe to reduce viscosity due to DNA breakage, followed by centrifugation at 13,000 rpm at room temperature for 15 min. Proteins were quantified by Pierce BCA Protein Assay Kit (Thermo Scientific, Waltham, MA, United States) and resolved by high resolution NuPAGE 4–12% Bis-Tris Gel (Life Technologies, Waltham, MA, United States) according to the manufacturer's instruction. Antibodies: Mouse α-NUDT21 (1:500, Santa Cruz Biotechnology, Dallas, TX, United States, sc-81109); rabbit α-serum MeCP2 N-terminus (1:5000, Zoghbi Lab, #0535); mouse α-GAPDH (1:10,000, Millipore, Billerica, MA, United States, CB1001-500UG).

## RNA extraction and quantitative real-time PCR

Lymphoblastoid cell suspension cell cultures were collected at $6 \times 10^6$ confluence and processed for RNA extraction. Total RNA was obtained using the miRNeasy kit (Qiagen, Netherlands) according to the manufacturer's instructions. RNA was quantified using the NanoDrop 1000 (Thermo Fisher, Waltham, MA, United States). Quality of RNA was assessed by gel electrophoresis. cDNA was synthesized using Quantitect Reverse Transcription kit (Qiagen, Netherlands) starting from 1 μg of DNase-treated RNA. Quantitative RT-polymerase chain reaction (qRT-PCR) experiments were performed using the CFX96 Touch Real-Time PCR Detection System (Bio-Rad Laboratories, Hercules, CA, United States) with PerfeCta SYBR Green FastMix, ROX (Quanta Biosciences, Gaithersburg, MD, United States). Real-time PCR results were analyzed using the comparative Ct method normalized against the housekeeping gene *GAPDH* (*Vandesompele et al., 2002*). The range of expression levels was determined by calculating the standard deviation of the ΔCt (*Pfaffl, 2001*). To ensure efficacy of the genomic DNA elimination, we ran negative control samples in the qRT-PCR that did not have reverse transcriptase (-RT) in the cDNA synthesis reaction.

## Northern blot assay

Total RNA was obtained from lymphoblastoid cell suspension cell cultures as mentioned above. Total RNA (15 μg) was separated on 1.2% formaldehyde agarose gel and transferred to Hybond-N+ nylon membrane (Amersham, United Kingdom), followed by UV crosslinking with Stratalinker 2400 (Agilent Technologies, Santa Clara, CA, United States). *MECP2* cDNA probe (1431 bp), which contains a majority of exon1 and exon2, was synthesized by random primed DNA labeling kit (Roche, France) according to the manufacturer's instructions. The blots were hybridized using ULTRAhyb hybridization buffer (Applied Biosystems, Waltham, MA, United States) according to the manufacturer's instructions. In brief, the blots were hybridized with radiolabeled probes ($10^6$ cpm/ml) in ULTRAhyb hybridization buffer for 2 hr at 68°C. The blots were then washed with 2× SSC, 0.05% SDS for 20 min at room temperature, followed by washing with 0.1× SSC, 0.1% SDS at 50°C for 10 min. The blots were exposed for 72 hr for analysis. For loading control, the blots were then stripped in boiling 0.2% SDS and hybridized with *GAPDH* probe.

## Polysomal profiling

$1 \times 10^6$ lymphoblastoid cells were treated with either 100 μg/ml cycloheximide (Sigma–Aldrich, St. Louis, MO, United States) for 60 min at 37°C before being washed with cold PBS containing 100 μg/ml Cycloheximide or Puromycin. The cells were then lysed in polysome lysis buffer (10 mM Tris-Cl, pH 7.4, 5 mM MgCl$_2$, 100 mM KCl, 1% (vol/vol) Triton X-100, 0.5% (wt/vol) deoxycholate, 1000 U/ml RNasin, 2 mM DTT and 100 μg/ml Cycloheximide in DEPC-treated water), incubated on ice for 10 min and centrifuged at 16,000×*g* at 4°C to obtain the clarified post-nuclear lysate. Post-nuclear clarified lysates (50 OD units) were layered on top of 10–50% sucrose gradients and centrifuged at 28,000 rpm (100,000×*g*) for 4 hr at 4°C using an SW41 rotor (Beckman, Fullerton, CA, United States). Sucrose gradients were fractionated using a BR-184 tube piercer and a syringe pump (Brandel, Gaithersburg, MD, United States) fitted with a UA-6 UV detector (Teledyne ISCO, Lincoln, NE, United States). Digital data were collected during fractionation using the DI-158U USB data acquisition device (DATAQ Instruments, Akron, OH, United States). The digital data were processed using the Peak Chart Data Acquisition Software and UV absorption at 254 nm was plotted vs time (from top to bottom or fraction number of sucrose gradients) (*Figure 3—figure supplement 2*). TRIzol LS reagent

(Life Technologies, Carlsbad, CA, United States) was used to extract RNA from the various polysome fraction and total lysate aliquots as per the manufacturer's recommendations.

## Primers

In order to unambiguously distinguish spliced cDNA from genomic DNA contamination, specific exon primers were designed to amplify across introns of the genes tested. The primers for all target genes tested were designed with Primer3 v.0.4.0 (*Koressaar and Remm, 2007*). Primer sequences:

### qRT-PCR

Human *GAPDH*: Forward, 5′-CGACCACTTTGTCAAGCTCA-3′; Reverse, 5′-TTACTCCTTGGAGGCCATGT-3′

Human *MECP2*: Forward, 5′-GATCAATCCCCAGGGAAAAGC-3′; Reverse, 5′-CCTCTCCCAGTTA CCGTGAAG-3′

Human long 3′ UTR: Forward, 5′-CATCCCGTGCTTTTAAGGAA-3′; Reverse, 5′-CATTCTCCCCAGC TCTTCAG-3′

Human short 3′ UTR: Forward, 5′-TCTGACAAAGCTTCCCGATT-3′; Reverse, 5′-CCAACTACTCC CACCCTGAA-3′

### Northern blot

Human *MECP2*: Forward, 5′-GAAGAAAAGTCAGAAGACCAGGACCT-3′; Reverse, 5′-GCTAACTCTC TCGGTCACGGG-3′

Human *GAPDH*: Forward, 5′-ACCACAGTCCATGCCATCAC-3′; Reverse, 5′-CACCACCCTGTTGC TGTAGCC-3′

## Acknowledgements

We are deeply indebted to the patients and their families for their willingness to participate in our research study. We would like to thank Pawel Stankiewicz for helpful contributions on patient data, Zhandong Liu from the Neurological Research Institute (NRI) Bioinformatics Core for his insight and expertise in interpreting the CLIP-seq data set, and Genevara Allen of the NRI Biostatistics Core for consultation and discussion. We thank Barbara G Cochran for critical reading of the manuscript. We would like to thank the Howard Hughes Medical Institute (HHMI–HYZ), the National Institutes of Health (NIH) (5R01NS057819) (HYZ), and the Rett Syndrome research trust (HYZ) for funding this research. The project was also supported by Baylor College of Medicine IDDRC Grant Number 1 U54 HD083092 from the Eunice Kennedy Shriver National Institute of Child Health & Human Development. The content is solely the responsibility of the authors and does not necessarily represent the official views of the Eunice Kennedy Shriver National Institute of Child Health & Human Development or the National Institutes of Health. CPS is generously supported by the Joan and Stanford Alexander Family.

## Additional information

### Competing interests

HYZ: Senior editor, *eLife*. The other authors declare that no competing interests exist.

### Funding

| Funder | Grant reference | Author |
| --- | --- | --- |
| Howard Hughes Medical Institute (HHMI) | | Huda Y Zoghbi |
| National Institutes of Health (NIH) | 5R01NS057819 | Huda Y Zoghbi |
| Baylor College of Medicine IDDRC | 1 U54 HD083092 | Huda Y Zoghbi |
| Rett Syndrome Research Trust | | Huda Y Zoghbi, Callison E Alcott |
| Joan and Stanford Alexander Family | | Christian P Schaaf |

The funders had no role in study design, data collection and interpretation, or the decision to submit the work for publication.

## Author contributions

HYZ, Conception and design, Analysis and interpretation of data, Drafting or revising the article; VAG, Design the experiments and Performed the experiments, Conception and design, Acquisition of data, Analysis and interpretation of data, Drafting or revising the article; CEA, Design the experiments and Performed the experiments, Acquisition of data, Analysis and interpretation of data, Drafting or revising the article; C-AC, MAG, Contributed to molecular work and lymphoblast cultures, Acquisition of data; AC, Performed the polyribosome experiments, Acquisition of data; JAR, EKR, JLS, SWC, SP, JWW, ERR, DDGH, Examined the patients and provided clinical data; WL, Shared CFIm25 data prior to publication, Contributed unpublished essential data; JRN, Analyzed the polyribosome experiments; CPS, Serves as communicating author for all clinical data, Conception and design, Analysis and interpretation of data, Drafting or revising the article

## Author ORCIDs

Jill A Rosenfeld, http://orcid.org/0000-0001-5664-7987

## Ethics

Human subjects: All individuals with NUDT21 copy number variants were enrolled in a research study approved by the Institutional Review Board of Baylor College of Medicine (H-25466). The consent form specifically allows for sharing of medical information and physical exam findings. All individuals whose lymphoblastoid cell lines were enrolled as controls had been enrolled in a research study approved by the Institutional Review Board of Baylor College of Medicine (H-25531) as unaffected sibling controls.

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
