## [Decision Letter]

[Editors’ note: a previous version of this study was rejected after peer review, but the authors submitted for reconsideration. The first decision letter after peer review is shown below.]

Thank you for choosing to send your work entitled “Individuals with *NUDT21* CNVs have neuropsychiatric disease and altered MeCP2 abundance via alternative polyadenylation” for consideration at *eLife*. Your full submission has been evaluated by Stylianos Antonarakis (Senior Editor) and three peer reviewers, one of whom is a member of our Board of Reviewing Editors. One of the three reviewers, Stephen Warren, has agreed to share his identity.

Based on our discussions, the individual reviews below and policy regarding submissions to *eLife* where additional significant experimentation is required, we regret to inform you that this submission of your work will not be considered further for publication. However, we would be willing to consider a substantially revised manuscript that includes additional experimentation to address all major concerns detailed below.

While all reviewers found the association between *NUDT21* CNVs and neuropsychiatric illness interesting and potentially important, enthusiasm was reduced based upon the small number of cases that were considered in detail, prior knowledge regarding the relationship between *NUDT21* dosage and expression of long and short forms of *MECP2* mRNA with specific implication of alternative polyA site utilization (Masamha et al., 2014). There were also specific experimental concerns about certain aspects of the functional studies in your manuscript. Foremost among these were the discordant performance of controls in the mRNA studies shown in Figure 2 which substantially influences data interpretation, and the lack of studies to specifically assess for a contribution of altered mRNA stability. Given that the CNVs under consideration span many genes, both Reviewers 1 and 2 suggested targeted complementation experiments in the deletion and/or duplication cells via recombinant expression and/or siRNA-based methods, respectively. Also both reviewers asked for more clinical information about the additional CNV patients where cell lines were not available to try to increase knowledge and confidence regarding the phenotypic spectrum associated with *NUDT21* CNVs.

Reviewer #1:

This interesting manuscript describes and characterizes the functional consequence of CNVs spanning *NUDT21* in patients with forms of intellectual disability (ID; 3 duplications and 1 deletion). *NUDT21* encodes CFIm25, a component of the heterotetrameric pre-messenger RNA cleavage factor I (CFIm) that is known to influence alternative polyadenylation (APA) site usage. Prior work suggests that increased CFIm25 dosage favors distal APA site usage for many transcripts, prominently including those for *MECP2*. This often (generally ?) associates with decreased protein expression, presumably due to the inclusion of additional negative regulatory 3'UTR elements such as those that encode binding sites for miRNAs that can function to decrease mRNA stability or (more commonly) translational efficiency. The reverse is true for reduced CFIm25 dosage. Importantly, either decreased or increased MeCP2 expression is known to cause ID with or without autistic features of variable severity.

Using patient lymphoblast lines, the authors show that *NUDT21* duplication or deletion CNVs associate with decreased or increased expression of MeCP2, respectively. Curiously, both types of CNVs correlate with increased *MECP2* mRNA expression. The authors describe apparent differences in APA site bias that might reconcile these findings and support their hypothesis that *NUDT21* CNVs represent a previously unrecognized etiology of ID.

In general, this is a quality study with very interesting and provocative findings. I do, however, have a number of questions and concerns that limit my enthusiasm for this manuscript in its present form. These are detailed below.

1) My major concern with this manuscript relates to the mRNA data presented in Figure 2. While the protein data nicely document increased CFIm25 and decreased MeCP2 protein expression in duplication patients, with the reverse in the deleted patient (n=1 concerns noted), the mRNA data are more ambiguous. Specifically, the northern blot data shown in 2D appear to show an increase in the short (proximal APA site usage) isoform in both the duplicated and deletion patient(s), although this is not specifically quantified. The duplication and deletion patient(s) appear to show increased or decreased distal APA site usage respectively, in keeping with hypothesis, but this is only true when each CNV type is correlated with their respective controls. The difference in the performance of the controls in each arm of this experiment is striking and without apparent basis, with essentially no usage of the distal APA site in the duplication blot and substantial (essentially that of the duplication patients) in the deletion blot. Perusal of the Methods suggests that this does not relate to exposure time or any other definable experimental parameter. If the deletion lane were compared to the controls for the duplication blot, then no apparent difference between deletion and duplication patients would be observed, leaving in question the molecular basis for the striking difference in protein expression between the groups. Even at face value, the (somewhat) higher total mRNA expression and the equivalent long/short isoform ratio shown for the deletion patient (when compared to controls) fails to fully explain the striking increase in MeCP2 protein expression that is evident from the blot in Figure 2 (and inadequately captured in the quantitative data shown below). It would be valuable to show a single northern blot with controls, the duplication patients and the deletion patient, with independent quantification of each parameter (long, short, total, ratio) to facilitate direct comparisons.

2) Given the authors’ original hypothesis that the long isoform would show lower stability, I am not sure that a simple measurement of steady state mRNA abundance adequately captures the potential complexity of the situation. It seems possible, even likely, that an attempt at transcriptional compensation is confounding the situation. In this light, it would be useful to measure mRNA half-life, given the authors’ stated goal to understand the mechanism of altered regulation of gene expression. This could be accomplished by measuring steady-state abundance of the various mRNA isoforms in close temporal sequence with administration of a transcriptional inhibitor such as actinomycin D or DRB. It seems possible that a higher than normal level of transcription might contribute to altered APA site usage in a manner at least partially independent from the absolute level of CFIm25, due to saturation of other trans factors.

3) The authors assume that observations made using lymphoblasts are representative of those in “all cells”. Is this necessarily the case? Could additional evidence be provided?

4) It is quite obvious that there are many genes in the described CNVs that might, in theory, contribute to MeCP2 and/or phenotypic expression. This might be addressed by attempts at compensatory overexpression of CFIm25 in cells from the deleted patient and/or siRNA-mediated knockdown in cells from the duplicated patients.

5) While it is stated that there is redundancy between CFIm59, CFIm68 and CFIm72, it is also stated that isolated deficiency of CFIm68 increases proximal APA site usage. In this light, it seems possible that CNVs that alter CFIm68 expression would also lead to relevant phenotypes. Has this been assessed?

6) The authors vaguely describe additional patients with CNVs encompassing *NUDT21*, but provide no additional details regarding phenotype, the nature of the CNVs (deletions or duplications), or their range. Despite the fact that cell lines were only available for the 4 described patients, this additional information could prove very relevant and informative. Also, have other CNVs in this region that do not involve *NUDT21* been associated with ID? Among patients with *NUDT21*-spanning CNVs and ID, were other obvious causes of phenotype identified?

Reviewer #2:

Gennarino et al. identify patients with large CNVs including *NUDT21* and show that MECP2 protein quantity correlates inversely with *NUDT21*, strongly linking dosage shifts to usage of alternate polyadenylation site in MeCP2 (i.e. *NUDT21* binds in complex to proximal site, leading to a longer 8.5 kbp 3'UTR, preferential degradation and reduced protein abundance). To test the relationship, the authors established lymphoblastoid lines from 3 duplication and 1 deletion patients and then assess MECP2 protein and RNA abundance site confirming the expectation.

The paper is clearly and concisely written. It is technically sound and the discovery of *NUDT21* is important in providing insight into the post-transcriptional regulation of *MECP2*, and of potential future therapeutic value with respect to Rett patients. It is a nice twist and use of CNVs to test specific hypotheses about gene networks.

The number of ID and DD patients is rather limited (3 duplication and 1 deletion); one of the subjects is mosaic (45%) drawing correlations is limited with such a low sample size. Because of its mosaic nature, duplication subject 3 should be indicated on Figure 2.

The relationship of high and low *NUDT21* expression in GBM tumors has already been shown to result in shifts in abundance of long and short *MECP2* transcripts presumably as a result of preferential PA site usage (see Figure 4, Masamha et al., 2014). The results presented here are not that novel or unexpected from a molecular perspective.

The CNVs are massive ranging in size from 1.8 Mbp to 18.1 Mbp affecting the dosage in many genes. Even the smallest region of overlap appears to involve more than a dozen genes (see Figure 1). How can the authors be so sure that other genes involved in polyadenylation site usage are not affected? Wouldn't a rescue experiment be more informative and specific (i.e. targeted knockdown of the *NUDT21* in duplication patients or increase expression of *NUDT21* in deletion patients).

The large size of the CNV limits clinical relevance at least with respect to the Rett phenotype. Why not report the phenotype of all 11 patients at least in general terms? I understand that only 4 patients consented for additional follow-up but isn't there even general properties of the clinical presentation that might provide additional insight?

The CNV burden is borderline significance (p=0.035). It appears that both deletion cases and duplication cases were lumped together in the burden test. It seems more appropriate to consider them separately.

Reviewer #3:

This manuscript from the Zoghbi laboratory presents a story that begins with published observation that pre-messenger RNA cleavage factor I subunit CFIm25 knockdown results in marked changes in *MECP2* protein levels and p(A) site usage. The authors cleverly propose that human mutations modulating the abundance of CFIm25 could alter *MECP2* levels leading to an abnormal phenotype as it is well established that *MECP2* levels are tightly controlled in humans. Accordingly, the authors searched clinical aCGH databases of ∼94,000 samples and identified 11 CNVs involving *NUDT21* which codes for CFIm25. Of the 11, 4 families were ascertained for follow-up studies. Using lymphoblastoid cells of three duplications and one deletion, the authors convincingly show an inverse correlation between *NUDT21* copy number and *MECP2* levels by altering p(A) usage sites, with the long 3'UTR showing low *MECP2* levels compared to the short 3-UTR. The authors make a reasonable hypothesis that the long 3'UTR has many more regulatory sites influencing translation than the short isoform.

This is a very well written paper with convincing data that represent one of the first post-transcriptional regulators of *MECP2*, which is notable for its need to be rigorously controlled. Moreover, it points to *NUDT21* mutations as potentially important causes of neuropsychiatric disease. It is therefore an important contribution.

---

## [Author Response]

[Editors’ note: the author responses to the first round of peer review follow.]

We are happy to report that we have been able to address the reviewers’ concerns and suggestions, which were quite useful in helping substantiate our hypothesis and strengthen the narrative. Briefly, we report more cases and provide a more detailed description of the phenotypic spectrum of patients with *NUDT21* CNVs. Notably, we now demonstrate that normalizing *NUDT21* in lymphoblastoid cells from *NUDT21* duplication patients restores MeCP2 levels to normal range. We also present a clearer molecular mechanism for how *NUDT21* regulates MeCP2 in patients with neuropsychiatric disorders. Performing a polyribosome fractionation assay on lymphoblastoid cells derived from *NUDT21* duplication patients, we discovered that the increased long isoform of *MECP2* is highly enriched in the untranslated fraction of lymphoblastoid cells.

*Reviewer #1*:

*1) My major concern with this manuscript relates to the mRNA data presented in*
Figure 2*. While the protein data nicely document increased CFIm25 and decreased MeCP2 protein expression in duplication patients, with the reverse in the deleted patient (n=1 concerns noted), the mRNA data are more ambiguous. Specifically, the northern blot data shown in 2D appear to show an increase in the short (proximal APA site usage) isoform in both the duplicated and deletion patient(s), although this is not specifically quantified. The duplication and deletion patient(s) appear to show increased or decreased distal APA site usage respectively, in keeping with hypothesis, but this is only true when each CNV type is correlated with their respective controls. The difference in the performance of the controls in each arm of this experiment is striking and without apparent basis, with essentially no usage of the distal APA site in the duplication blot and substantial (essentially that of the duplication patients) in the deletion blot. Perusal of the Methods suggests that this does not relate to exposure time or any other definable experimental parameter. If the deletion lane were compared to the controls for the duplication blot, then no apparent difference between deletion and duplication patients would be observed, leaving in question the molecular basis for the striking difference in protein expression between the groups. Even at face value, the (somewhat) higher total mRNA expression and the equivalent long/short isoform ratio shown for the deletion patient (when compared to controls) fails to fully explain the striking increase in MeCP2 protein expression that is evident from the blot in*
Figure 2
*(and inadequately captured in the quantitative data shown below). It would be valuable to show a single northern blot with controls, the duplication patients and the deletion patient, with independent quantification of each parameter (long, short, total, ratio) to facilitate direct comparisons*.

This point is well taken. We agree with the reviewer that it would be better to show a single northern blot image with controls, duplications, and deletions all together. We did the experiment and have included that image (Figure 4), which replicates our previous findings. However, we want to emphasize that the quantification presented in the first manuscript was not exclusively of the image shown, but from all three experiments performed. Importantly, we did not compare the CNV data with the controls on the blot alone, but with all the controls in every experiment, as indicated in the figure legend. We presented the northern blot quantification as a ratio of the long and short *MECP2* isoforms in the original text because the isoform switching was most pertinent to our narrative. However, we have included the independent quantification of each isoform in Figure 4. It has proven challenging to capture both the short and long isoforms on the same blot due to differing transfer efficiencies and required exposure times. We therefore decided to move the northern blot to the supplemental materials. We believe that the qPCR data are more compelling, and the new insight gained from the polyribosome assay nicely explains the fate of the RNA

Author response image 1.Northern blot assay from patient-derived lymphoblast cells.Northern blot assay showing that duplication and deletion patients respectively have more long or short *MECP2* isoforms (left panel). Quantification of three duplication patients and one deletion compared to 7 age-matched controls (right panel). Data represent mean ± SEM. The p values were calculated by Student's *t*-test comparing controls with duplication patients. **p*<0.05. M=mosaic patient.**DOI:**
http://dx.doi.org/10.7554/eLife.10782.012

*2) Given the authors’ original hypothesis that the long isoform would show lower stability, I am not sure that a simple measurement of steady state mRNA abundance adequately captures the potential complexity of the situation. It seems possible, even likely, that an attempt at transcriptional compensation is confounding the situation. In this light, it would be useful to measure mRNA half-life, given the authors’ stated goal to understand the mechanism of altered regulation of gene expression. This could be accomplished by measuring steady-state abundance of the various mRNA isoforms in close temporal sequence with administration of a transcriptional inhibitor such as actinomycin D or DRB. It seems possible that a higher than normal level of transcription might contribute to altered APA site usage in a manner at least partially independent from the absolute level of CFIm25, due to saturation of other trans factors*.

As recommended, we performed an RNA stability assay and found that the long *MECP2* isoform has a slightly longer half-life (5.8 hours) compared to total *MECP2* (5.1 hours) in patient-derived lymphoblastoid cells. This could partially account for the elevated *MECP2* in the duplication patients due to their elevated levels of long *MECP2*, but we cannot exclude other possibilities such as a feedback mechanism. Given the small difference in the half-life, and that it does not change any of the paper’s conclusions, we have decided not to include it in the revised text. As mentioned above, we believe the ribosomal data are more critical to the conclusions of the paper.

*3) The authors assume that observations made using lymphoblasts are representative of those in “all cells”. Is this necessarily the case? Could additional evidence be provided*?

Patient-derived lymphoblastoid cells are often used for patient-specific molecular and functional studies. Gene expression in lymphoblastoid cells covers a wide range of metabolic pathways in common with neurons. While they may not be representative of “all cells”, they represent an important model system, due to the minimal invasiveness of obtaining them, and the ease of cell maintenance. See Sie L. et al., J Neurosci Res., 2009 (PMID:19224581) for a review.

*4) It is quite obvious that there are many genes in the described CNVs that might, in theory, contribute to MeCP2 and/or phenotypic expression. This might be addressed by attempts at compensatory overexpression of CFIm25 in cells from the deleted patient and/or siRNA-mediated knockdown in cells from the duplicated patients*.

We thank the reviewer for this suggestion. We first looked for patients with point mutations in *NUDT21* or smaller CNVs and were able to identify one with a duplication of only 700 kb, reducing our minimal region to just five genes. Additionally, while it is not standard to transfect patient-derived lymphoblastoid due to low efficiency and cell survival following transfection, we tried a novel approach using nucleofection, and were successful. We adapted a T-cell protocol from Lonza, and as can be seen in Figure 2 and Figure 2—figure supplement 1, nucleofection of anti-*NUDT21* siRNA in control lymphoblastoid cells decreases NUDT21 while increasing MeCP2 as in the deletion patient. More importantly, normalizing NUDT21 levels in the *NUDT21* duplication patient lymphoblastoid cells restores MeCP2 levels to the normal range. These compelling data prove that NUDT21 regulates MeCP2 in our patients.

*5) While it is stated that there is redundancy between CFIm59, CFIm68 and CFIm72, it is also stated that isolated deficiency of CFIm68 increases proximal APA site usage. In this light, it seems possible that CNVs that alter CFIm68 expression would also lead to relevant phenotypes*. *Has this been assessed*?

We have found patients with CNVs spanning *CPSF7* (which encodes CFIm59) and *CPSF6* (which encodes CFIm68) with phenotypes similar to those of the *NUDT21* CNV patients, and we have contacted their physicians for clinical data and blood samples. However, it took 15-18 months to receive the samples for the *NUDT21* CNV patients, and will take a comparable amount of time to receive those. Thus, we will happily follow up with a Research Advance when that study is complete.

*6) The authors vaguely describe additional patients with CNVs encompassing* NUDT21*, but provide no additional details regarding phenotype, the nature of the CNVs (deletions or duplications), or their range. Despite the fact that cell lines were only available for the 4 described patients, this additional information could prove very relevant and informative. Also, have other CNVs in this region that do not involve* NUDT21 *been associated with ID? Among patients with* NUDT21*-spanning CNVs and ID, were other obvious causes of phenotype identified*?

We were able to obtain a detailed clinical history for one additional duplication case, which we now include in the manuscript. We also described the remaining CNVs by type, as the reviewer suggested, in the revised manuscript. However, we were unable to obtain clinical information beyond the indication for aCGH testing available in the databases because the primary physician did not respond to our queries. As we now mention in the revised text, none of the other genes in the minimal region of overlap are associated with neurological disease when heterozygous. Additionally, there are no individuals in the patient databases with CNVs of adjacent genes that do not also span *NUDT21*. All this together, along with the rescue experiment in the duplication lines, supports our conclusion that *NUDT21* alters MeCP2 levels, and, hence, is the driver of pathogenicity.

*Reviewer #2*:

*[…] The number of ID and DD patients is rather limited (3 duplication and 1 deletion); one of the subjects is mosaic (45%) drawing correlations is limited with such a low sample size. Because of its mosaic nature, duplication subject 3 should be indicated on*
Figure 2.

We have labeled the mosaic patient on the figures. We thank the reviewer for the helpful suggestion.

*The relationship of high and low* NUDT21 *expression in GBM tumors has already been shown to result in shifts in abundance of long and short* MECP2 *transcripts presumably as a result of preferential PA site usage (see Figure 4, Masamha et al., 2014). The results presented here are not that novel or unexpected from a molecular perspective*.

It is true that the inverse relationship between *NUDT21* and MeCP2 via polyadenylation has been established in tumor cells; however, we make several novel contributions. We identify *NUDT21* as a candidate intellectual disability and autism gene, and show that patients with *NUDT21* CNVs have altered MeCP2. Importantly, in this manuscript we elucidate the mechanisms by which *NUDT21* regulates MeCP2, i.e. by increasing the long mRNA isoform, which we then show is inefficiently translated.

*The CNVs are massive ranging in size from 1.8 Mbp to 18.1 Mbp affecting the dosage in many genes. Even the smallest region of overlap appears to involve more than a dozen genes (see*
Figure 1*). How can the authors be so sure that other genes involved in polyadenylation site usage are not affected? Wouldn't a rescue experiment be more informative and specific (i.e. targeted knockdown of the* NUDT21 *in duplication patients or increase expression of* NUDT21 *in deletion patients)*.

We thank the reviewer for suggesting this experiment. As mentioned above in response to Reviewer 1, it is not common to transfect lymphoblastoid cells to target a putative causal gene in CNV patients due to low transfection efficiency and survival of lymphoblastoid cells upon transfection. However, we attempted the experiment by optimizing a Lonza nucleofection protocol originally designed for T-cells. We are happy to report, as shown in Figure 2 and Figure 2—figure supplement 1, that we succeeded in this experiment. We nucleofected lymphoblastoid cells from control and *NUDT21* duplication patients with small interfering RNA against *NUDT21* and found that decreasing the *NUDT21* levels in control patients increases MeCP2 like in the deletion patient, while normalizing its level in duplication patients completely restores MeCP2 protein levels. Because we could not find examples of successful transfections of patient-derived lymphoblastoid cells in the literature, we have provided a detailed protocol.

*The large size of the CNV limits clinical relevance at least with respect to the Rett phenotype. Why not report the phenotype of all 11 patients at least in general terms? I understand that only 4 patients consented for additional follow-up but isn't there even general properties of the clinical presentation that might provide additional insight*?

We were able to obtain a detailed clinical history from one additional patient with a duplication of only 700 kb. However, all we know about the remaining patients is their indication for aCGH testing, which is incomplete. We have put as much information in the manuscript as we could successfully obtain from referring physicians.

*The CNV burden is borderline significance (p=0.035). It appears that both deletion cases and duplication cases were lumped together in the burden test. It seems more appropriate to consider them separately*.

To support our hypothesis that *NUDT21* dosage change leads to neuropsychiatric disease, we believe it is unnecessary to separate the duplications and deletions in the burden test, because we are showing that the brain is sensitive to the level of *NUDT21* such that both increases and decreases lead to neuropsychiatric disease. Additionally, since Signature Genomics is no longer in business, we cannot sort their database to get precise numbers for their total duplication and deletion cases. We also reduced the number of patients in our denominator by 6,300 by excluding prenatal cases for whom we could not know if they would have had neuropsychiatric disease.